# Comparative Study in Physical Fitness in Recreative Young Padel Players

**DOI:** 10.3390/jfmk10020214

**Published:** 2025-06-05

**Authors:** Ana Pereira, Luis Leitão, Diogo L. Marques, Daniel A. Marinho, Henrique P. Neiva

**Affiliations:** 1Instituto Politécnico de Setúbal, Escola Superior de Educação, 2910-761 Setúbal, Portugal; luis.leitao@ese.ips.pt; 2Sport Physical Activity and Health Research & Innovation Center, 2040-413 Rio Maior, Portugal; 3Quality of Life Research Centre, Polytechnic Institute of Setúbal, 2910-761 Setúbal, Portugal; 4Department of Sport Sciences, University of Beira Interior (UBI), 6201-001 Covilhã, Portugal; diogoluis.sequeira@gmail.com (D.L.M.); dmarinho@ubi.pt (D.A.M.); hpn@ubi.pt (H.P.N.); 5Research Center in Sport Sciences, Health Sciences and Human Development (CIDESD), 6201-001 Covilhã, Portugal

**Keywords:** youth, exercise, power, training

## Abstract

**Background:** In recent years, padel-based interventions have been widely applied in junior and elite players of both genders concerning athletic performance, whereas evidence of their efficacy in trials that use simple randomization has not been well established. This study aimed to compare the effects of 8 weeks of padel training (PD) on the strength and power of untrained healthy children. **Methods:** Twenty-five children aged 11–15 years (12.36 ± 1.15 years) were randomly assigned into experimental (PD) (nine boys and five girls: 1.58 ± 0.04 m; 50.00 ± 6.75 kg; and 19.96 ± 1.95 kg/m^2^) and control (CT) (seven boys and four girls: 1.60 ± 0.05 m; 56.92 ± 2.75 kg; and 21.61 ± 1.02 kg/m^2^) groups. The PD group trained twice a week for 8 weeks, and the CT group did not follow any training program and did not participate in regular exercise or sports. Countermovement jump, medicine ball throw, handgrip, and 5 m sprint test results were measured at baseline and after the intervention in the padel sport group. **Results:** The results showed a significant interaction for training-induced responses in the sprint test (T5) (F = 10.55, *p* = 0.004, η^2^ = 0.31). No significant interactions were observed for handgrip strength (HG) (F = 3.90, p=0.06), the medicine ball throw (MBT) (F = 0.851, *p* = 0.37, η^2^ = 0.04), and the countermovement jump (F = 1.04, *p* =0.32, η^2^ = 0.04), with clear improvements from pre- to post-training in the PD group. After 8 weeks of training, the PD group showed increased performance in handgrip strength (*p* = 0.004), while the CT group had decreased velocity post-training (*p* = 0.011). **Conclusions:** The individual results in the PD group showed an improvement, which suggests that the practice of padel seems to be a good strategy for improving one’s fitness. It is suggested that 8 weeks of PD seem to be effective in improving strength- and power-related variables in healthy, untrained children. This could be considered an alternative to traditional sports to improve the body fitness of young children and should be applied in school-based programs and the sports club community. Also, more high-quality RCTs are needed in the future.

## 1. Introduction

Padel is a relatively new and high-intensity sport with a steady increase in participation over the past 10 years. It has been growing in popularity, highlighting the need to expand our understanding of it [1]. Padel is characterized by some explosive movements, the need to change directions, and racket skills. It is played in pairs on a 20 × 10 m court surrounded by glass walls and a metal fence on which the balls can bounce [2,3,4]. The main physiological characteristics of its players (namely, oxygen consumption, mean heart rate, and perceived exertion rates) are like those of other racket sports. The alternating intervals between high- and moderate/low-intensity exercise are above 50% of players’ VO_2_ and 74% of players’ maximum HR [2,3]. It has become one of the most popular sports worldwide, namely in Spain, Argentina, Mexico, Sweden, the United Arab Emirates, and the United Kingdom. It includes players of all genders, ages, and fitness levels [5]. Padel’s popularity has grown exponentially, becoming one of the most practiced sports in Portugal [6]. The number of licenses has increased in the last 2 years, particularly licenses for young players (younger than 18 years old), resulting in many players at the Portuguese Youth National Padel Championship [7]. This evolution, however, has not been accompanied by adequate developments in research on the physical activity demands on young padel players during matches [5,8]. In a padel match, each player must have elevated muscular and strength performance with an optimal response to fast movements and when using the upper and lower limbs [9,10]. Each match is characterized by intermittent whole-body efforts, with many strokes, repeated bouts of high-speed running, and changes of direction between accelerations and decelerations [2,4,5].

Therefore, in junior and senior players, physical fitness is important to prevent injuries [11,12] and to factor in body limitations according to players’ competitive level [13]. Padel has many benefits, and recent studies have shown significant results in terms of body composition, fitness capacity at the junior and elite levels for females and males [14], and some psychosocial characteristics, such as increased self-performance in young padel players of both genders between 13 and 16 years old [15]. However, the literature concerning the analysis of the physical fitness of young players during formative stages is scarce compared to other sports [16]. Several factors could have an influence on young players of both genders [17], and recent studies have shown that experience could interfere with some variables, namely in explosive and strength tests [18], reporting that male players between 11 and 16 years old have greater strength and speed than females [19,20]. This could be related to gender differences in anthropometrics, especially that of the appendicular muscle, and the physiological characteristics of players in this age group. This could affect one’s actions in a match, namely one’s acute responses to intensity, recovery, speed of actions using the lower and upper limbs, such as smashes and backhand volleys, and hydration status.

Analyzing the physical level of young padel players compared with a young population that does not play sports could improve and amplify the importance of padel sports in schools and communities. Since 2021, there has been a great increase in female participation in this sport [21]. Moreover, it seems to lead to an increase in physical activity at this age, which plays an important role in maintaining a healthy body composition, strength, and balance [15]. 

In addition, examining the effect of training processes during puberty could help sports trainers and teachers identify asymmetries between the upper and lower limbs and prevent possible injuries and muscular decompensations [11,15]. In the upper limbs, great ranges of motion during overhead strokes can increase the risk of shoulder, elbow, and wrist lesions [22,23]. Similarly, in the lower limbs, repetitive concentric and eccentric loads, as well as jumps, appear to increase injuries in the knees, legs, and feet [23]. The trunk region in young padel participants with abdominal wall muscle injuries can also contribute to musculoskeletal injuries. A detailed understanding of the biomechanical features of padel-specific gestures is fundamental to prevent and promote performance [23]. Furthermore, analyzing the effect of training, following muscular performance and physiological measures at these ages, and understanding the interactions between players’ performance and health could be useful for clubs, associations, and federations. Additionally, international data from different scientific areas reveal that lean mass, namely muscular mass, is the major contributor to general health [12,24], preventing aging diseases and promoting longevity. So, a better understanding of padel physical performance in youth athletes would enhance information accuracy on muscular characteristics in the formative stages of racket sports participants, including the inclusion of padel in curricular programs at schools.

Due to the importance of analyzing the impact of physical level on young participants in padel, we hypothesize that 8 weeks of padel training will result in significant improvements in strength and speed compared to the control group. Therefore, this study aimed to assess the physical fitness values in upper- and lower-body strength in young padel players between 11 and 15 years, both male and female, and to determine the possible differences compared to children who do not engage in any sports activities.

## 2. Materials and Methods

### 2.1. Participants

The present study was conducted on a target population of leisure participants under 11 and under 15 years old in a community in Portugal. A total sample size of sixteen participants would be required to achieve a power of 80%. This assumes an alpha level of 0.05, a correlation among repeated measures of 0.7, a medium effect size (ηp2 = 0.09), two groups, and two assessments (pre- and post-test). Calculations were performed using a specific software (GPower, v.3.1.9, University of Kiel, Kiel, Germany). Participants of different classes in the north of Portugal were randomly chosen and were informed about the study protocol, risks, and benefits, and once they agreed, they voluntarily signed the informed consent form, in addition to the form being signed by their parents or responsible family member.

All the participants had to meet the following inclusion criteria: (i) to be between 11 and 15 years old; (ii) to not to have any injury or illness during the investigation or at least six months before the study; (iii) to not have padel experience; and (iv) to practice only the sport of padel. Subjects were excluded from the study if they presented a recent hospitalization, severe cognitive or motor impairments, an inability to exercise, and any other medical contra-indications for physical exercise. The experimental group performed PD training twice weekly for 8 consecutive weeks, and the CG maintained their daily routine. The participants were recruited one week before the beginning of the evaluation and after 8 weeks of training. The first training session was the pre-test, and the last training session was the post-test. In the same week, it was performed for the control group.

In each session of padel training, the participants performed a warm-up with 15 min dynamic exercises between low and moderate intensity (self-perceived) consisting of jogging with velocity changes, mobility, jumps, and hitting racquet balls in pairs.

Before the data collection, the participants were informed about the study procedures and signed an informed consent that was written by the researcher who conducted and designed the study. Participants were advised to maintain their previous lifestyle throughout the study, including dietary patterns and daily routines. The procedures were performed according to the Declaration of Helsinki and were approved by the scientific board of Sport Science of the University of Beira Interior, Portugal.

### 2.2. Procedures

The present study consisted of a non-randomized controlled trial that aimed to verify the changes in physical fitness in muscular strength of the lower and upper limbs after 8 weeks of padel training. These variables were assessed during the week before the padel training classes started (week 0—pre-training) and the week after the end of the program (week 9—post-training). The variables assessed could be categorized into different groups, namely, (i) anthropometry, with the measurement of height, weight, and body mass, and (ii) physical condition, through the evaluation of the explosive strength of the upper limbs (1 kg medicine ball throwing), maximal strength (handgrip strength), and strength of the lower limbs (countermovement jump and running speed). All tests were performed in this order: body composition, countermovement jump, ball throwing, running speed, and handgrip strength.

Before the assessments, the sample underwent a familiarization session with all the instruments and evaluation tests. The participants and their coaches were requested to abstain from intense activities for 48 h before the assessments. The assessments were always performed at the same time (6 h 30 p.m.) and under the same environmental conditions (~20 C, ~60% humidity). Before the familiarization session and assessments, a standardized 15 min warm-up based on general mobility and low-intensity continuous running was performed. Participants rested 3–5 min between each test.

#### 2.2.1. Anthropometric Measures

All anthropometric variables were performed by the same experienced evaluator. Stature was measured to the nearest 0.1 cm using a stadiometer, and body mass was measured to the nearest 0.1 kg using a portable scale (model 707, Seca Corporation, Columbia, MD, USA) in nude, barefoot conditions. Body mass index (BMI) values were obtained from the above parameters [21].

#### 2.2.2. Handgrip Strength Measurement

A grip strength dynamometer (Takei Kiki Kogyo, Tokyo, Japan) was used to determine handgrip strength (HG) in both the right and left hands. The dynamometer was adjusted for each participant’s hand size. The participants maintained a standing position with the shoulder adducted and neutrally rotated and the elbow fully extended. The dynamometer was held freely without support, not touching the subject’s trunk. The participants were instructed to perform a maximal isometric contraction for five seconds. Each participant completed three trials with each hand, with a 1 min rest between trials, and the highest scores (in kg) were recorded [25].

#### 2.2.3. Medicine Ball Throwing Measurement

Medicine ball throwing performance was tested with a 1 kg medicine ball (MBT 1-kg). Each subject sat on the floor with the posterior trunk region positioned against the wall and held the ball to the front with both hands. Three approved attempts were made with 1 min rest intervals. The maximal throwing distance was determined using a flexible steel tape. Only the best attempt was used for further analysis [25].

#### 2.2.4. Speed Test

Five meters (T5m) is most specific to the distance covered in any one effort during a padel match. Outside of a padel court, the participants were required to cover such distance in the shortest time they could. Each individual began in a stationary position. The time (in seconds) to run 5 m was obtained using photocells (Brower Timing System, Fairlee, VT, USA). Three trials were performed, and the best time scored (seconds and hundredth) was registered [26].

#### 2.2.5. Vertical Jump Measurement

Before a warm-up consisting of several submaximal jumps, the participants performed the evaluation of lower-body muscular strength assessed using the Infrared Platform Ergo Jump Plus-Bosco System (Bmedic, S.C.P., Barcelona, Spain). The countermovement jump (CMJump) test was selected due to its high reliability [25]. Each participant completed three maximal countermovement jumps with 3 min of rest between trials. The best score was recorded. All CMJumps were completed by keeping the hands on the hips throughout the test. Whilst standing erect, participants were instructed to flex their knees into a squat position (90) and then immediately rebound in a maximal vertical jump as high as possible. No pause was allowed between the eccentric and concentric phases, and participants landed with both feet in contact with the floor. The measured height was expressed in centimeters and was converted to power (w) [27].

### 2.3. Statistical Analysis

Data are expressed as means and standard deviation. Statistical data were analyzed using Microsoft Office Excel^®^ (Microsoft Inc., Redmond, WA, USA) and SPSS v28 (SPSS Inc., Chicago, IL, USA). The figures were designed in GraphPad Prism v7 (GraphPad Inc., San Diego, CA, USA). A value of *p* < 0.05 was considered statistically significant [28]. To test normality, the Shapiro–Wilk test (n < 30) was used, followed by the Levene test to confirm the homogeneity of variances. The independent sample *t*-test was used to compare pre-training variables between groups, allowing for the determination of any significant differences at baseline between groups. A two-way mixed-design analysis of variance (ANOVA) with one factor with repeated measures (pre-test and post-test) was used to examine how different interventions affect outcomes over time, assessing both the within-subject effects (changes over time) and between-subject effects (differences between groups). An independent samples *t*-test compared the differences between groups in the percent change from pre- to post-training ([post-session − pre-session]/pre-session) × 100). We also calculated the effect size to estimate the variance between groups, using the partial eta squared (ηp2), and Hegde’s g (g) was calculated for within-subject comparisons. The effect size (g) was interpreted as follows: trivial, 0.0–0.2; small, 0.2–0.6; moderate, 0.6–1.2; large, 1.2–2.0; very large, 2.0–4.0; and extremely large, 4.0 [14]. The ηp2 values were classified as small (0.01–0.08), moderate (0.09–0.24), and large (≥0.25).

## 3. Results

An initial sample of 29 individuals was assessed for eligibility, and 25 agreed to take part in the study upon meeting the criteria for selection, with two weeks of recruitment before the collected data. The twenty-five children aged 11–15 years (mean SD: 12.36 1.15 years) were recruited into PD (nine boys and five girls) and control (CT) (seven boys and four girls; no training) groups (Table 1). Table 2 shows the results of handgrip strength, medicine ball throwing, 5 m sprint running, and countermovement jump. At baseline, no significant differences were found between groups regarding age, body mass, height, and BMI (*p* ≥ 0.001). Significant group interactions between pre- and post-training were observed only for T5 (F 1.3 = 10.55, *p* = 0.004, n^2^ *p* = 0.31) with large effects. No significant interactions were observed for HG (F 1,23 = 3.90, *p* = 0.06), BM (F 1.23 = 0.851, *p* = 0.37, n^2^ *p* = 0.04), and CMJump (F 1.23 = 1.04, *p* = 0.32, n^2^ *p* = 0.04). There were differences on the CMJump test, with PD jumping ~2–3 cm higher. Practice experience might have influenced the performance in the MBT for both groups. Also, jump disparities may not occur due to the practice of padel but because of natural physical differences at these stages. In the PD group, handgrip was improved, with 3kg more strength. Nevertheless, sprint time was the main difference between the groups, with the control group showing a substantial increase in time to perform 5 m~2 s more. When analyzing each group from pre- to post-training, the PD group showed improvements in handgrip strength (*p* = 0.004, g = 1.44), while the CT group showed decreased sprint performance (*p* = 0.01, g =0.87). These differences can be observed by analyzing the effect size values presented in Figure 1. Lower magnitudes of the effects were found in the CT group among the different variables. Medium-to-large effects were found in the PD group, with higher values obtained in the handgrip and sprint test.

Figure 1 presents the pre- and post-training changes, underlying the previous results of significant differences between groups in the sprint performance. In the experimental group (PD), 29% of participants improved CMJump (n = 4), 79% improved medicine ball throw (n = 11), 86% improved handgrip strength (n = 12), and 71% improved sprint performance (n = 10). In the control group, 63% of participants improved CMJump (n = 7), 36% improved medicine ball throw (n = 4), 45% improved handgrip strength (n = 5), and 9% improved sprint performance (n = 1).

## 4. Discussion

### 4.1. Summary of Evidence

The present study provides new data concerning physical fitness in recreative young padel players and its evolution after 8 weeks of training. Concerning the results, this study provides novel insight into the muscular performance of padel in young players, namely, for handgrip strength, explosive strength of the lower extremities, and running speed [15,29]. In the present study, our results suggest that velocity requirements that are related to specific padel game characteristics and skills must be developed at these ages because this could help padel performance in the future and prevent injuries [30]. Upper- and lower-limb strength in our study was markedly lower when compared to other racket sports, characterized by a poorer throwing and jumping ability in players aged 11 to 16 [9,14]. These results suggest that the weaker strength and conditioning profile in young padel players still needs exploration.

Padel is fundamentally unilateral in the upper limb and uses repetitive movements. In this case, with young players, all the structural, muscular, and functional developments should be addressed during the training process, even in leisure classes. More studies should research symmetry and fatigue. In a recent study developed by Viera et al. (2025), it was shown that professional Brazilian padel players presented reduced rotation and torque in the shoulder [31]. In this case, according to recent studies [11,30,31,32], resistance training that focuses on the development of strength in the upper limbs should be promoted. Regarding all the technical skills used in padel with a racket, kinematic analyses could contribute to understanding and correcting the execution of the athletic gesture [33]. Strength training must be a mediator between the rehabilitation plan and the athlete’s needs.

After 8 weeks of training, running speed was the only variable that had a significant impact between the groups. Indeed, muscular performance comparisons showed that the padel group between the pre-training and the post-training increased handgrip strength and also speed. Once again, as an intermittent sport, padel seems to be characterized by repetitions of rapid starts where muscular performance in the upper and lower limbs is required [20]. Above these positive results, in the control group during the same period, we observed a significant decrease in speed performance over 5 m. The increase in handgrip strength and speed may be associated with racket use and specific training in the court game [29]. However, above any sports participation by the control group, we must be concerned about the decrease in lower limb speed, as it can lead to various health and physical consequences. Compared to different sports disciplines, a recent study by Kramer et al. [33], observed that sprint performance in young female tennis players appeared to depend on maturation and growth in parallel with physical fitness until 14 years. The same authors found that in males, sprint performance was related to longitudinal changes in body size and lower limb strength up until age 13 [34]. Thus, in the present study, our results in BMI between both groups could interfere with the results in sprint performance, and this must be considered in future studies.

Muscle strength is a crucial aspect of children’s development [9,26]. Specifically, in the lower limbs, muscular performance contributes to locomotive movement and is important for play and social interaction [17,24]. Without adequate strength, young individuals may experience reduced physical engagement and social interaction, which can be associated with more indoor activity with electronic equipment [29]. This focus on physical activity and muscular strength in today’s youth is a priority. Regular physical activity of adequate intensity and quality is crucial for developing sufficient muscle mass to handle the demands of daily activities throughout life [29]. Concerning handgrip strength in our study, we must spotlight the results achieved in handgrip performance in padel players [14,35]. Reduced handgrip strength is related to cardiometabolic risk factors, neuromuscular diseases, progressive developmental diseases, and children involved in accidents [36]. In the control group, a plausible explanation for the lower handgrip strength may also be the actual absence of physical activity in the present sample.

The padel game structure is characterized by a higher amount of ball hits in comparison with other racket sports (tennis, badminton, and squash) [14,36]. So, the strength used by the dominant hand is naturally improved. In addition, the offensive aim in padel should be playing near the net to improve the probability of winning the match [37]. For this purpose, players use specific defensive strategies, mostly through lobs, to send the opponents back to the baseline and recover the net [3]. These movements appear to be a tactical response to avoid the opponents reaching the net and, in this case, the players needed to run faster. Padel players require high power in their lower limbs to quickly shift their upper-body weight. In our study, this was observed by the increase in performance in handgrip strength and the 5 m test. In the future, it is necessary to understand the resting time of each player during power exercises that could decrease individual performance by the increase in fatigue. This may improve the quality and specificity of the training process, resulting in greater application and transfer to the competition in padel.

Following recent research that examined fitness levels in young padel players, which studied boys and girls aged 11 to 16 years [5], it was identified that there were gender differences, with boys performing faster 10 m and 20 m sprints and having greater throwing strength (overhead medicine ball throw) than girls. However, considering the small size of the padel court and the fact that the players mostly made diagonal or lateral movements [16] and covered less than 8 m running [38], it suggests the relevance of short running bouts. However, in the present study, we define 5 m running, which merits further examination. Similarly, in future studies the assessment of swinging actions by rotational medicine ball throws would be advisable to describe strength performance in padel [39]. Nevertheless, we observed a difference in body mass index (BMI) between the two groups, and considering the age of our participants, those in the experimental group could be classified within a normal BMI category, whereas those in the control group might fall into an overweight category. This discrepancy could potentially influence the manifestation of other physical fitness variables.

Our findings highlight the need for additional research regarding longitudinal studies to observe the evolution of fitness status from young to older ages and to confirm whether adherence to padel practice in childhood could lead to a healthier adulthood.

### 4.2. Limitations

The present study presents some limitations that should be taken into account in future studies to complement the reduced number of tests that we used in our study. Thus, according to our study population, the following aspects could be improved upon: (i) it would be of great interest to observe the difference between boys and girls with a larger sample; (ii) padel-specific tests have not been undertaken for young people or even used in racket sports; (iii) the results presented have validity compared with the control group, but it would be interesting to analyze the results between different sports that are practiced in this age range and at a larger scale; and (iv) concerning the padel game, the individual side of each player in the court should be signalized to analyze other physical and psychomotor qualities and abilities, such as lactate levels, maximal oxygen consumption, hematological parameters, mental fatigue, and padel accuracy (i.e., drive, drive volley, bandeja, and drive attack after the use of the glass). It is essential to improve this research to contribute to padel dissemination in schools, considering its capacity to improve physical activity and promote general health. In addition, obtaining other muscular parameters, such as muscle percentage in upper and lower limbs and adiposity, may contribute to understanding the effect of this sport on different ages and prevent discrepancies between genders that may require different types of training. In our study, the body mass index was different between groups, and as we did not measure muscle mass, this could have induced changes in performance. In this case, it will also be interesting to assess dietary control. Nevertheless, the present study showed the strengths of practicing padel in young people that overcome its limitations.

### 4.3. Practical Applications

The information provided in this study may contribute to planning specific training sessions for young padel players according to competitive demands characterized by padel game requirements. It is recommended that coaches and trainers include preventive interventions in padel players to avoid decrements in performance or the early abandonment of practice due to suffering an injury, particularly in the formative stages at young ages. In addition, considering that Physical Education must be one of the fundamental elements for health promotion in the school population, teachers are encouraged to use padel as an easy-to-learn and enjoyable sport to engage youths in physical activity. This could be complementary to other curricular sports, once per week, with an option to integrate two more weekly training sessions in the sports club of the school for students that want to achieve a better performance or in a padel club with a sports trainer. In the future, we must focus more on the validity and applicability of the trial findings.

## 5. Conclusions

According to the results, in the present study, young padel players showed better results in handgrip strength and 5 m sprint running, while the group that did not perform any sport showed a decrease of 6% after 8 weeks, specifically in velocity running. Due to the young age of the participants and the scarce information on the assessment of muscular components, these results could be used as a reference by padel coaches/physical trainers considering the maturational stage of the participants between 11 and 15 years. Nevertheless, our study also contributes to scientific evidence that the padel sport can be a good strategy to introduce new skills at schools and improve physical activity and performance in young participants.

## Figures and Tables

**Figure 1 jfmk-10-00214-f001:**
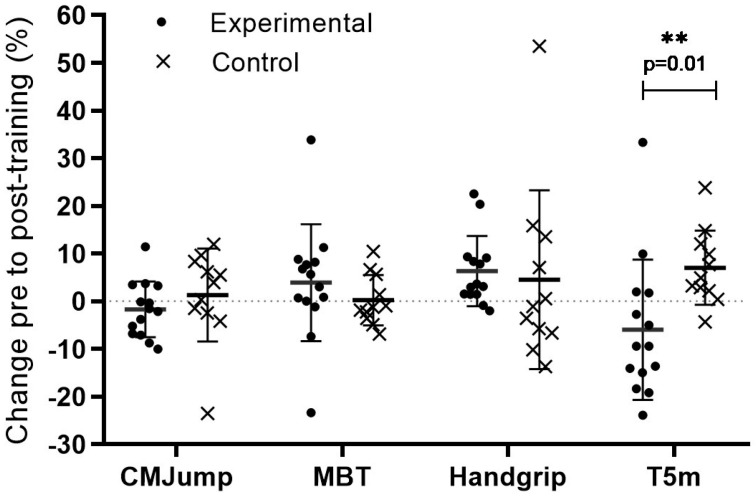
Individual value plot of changes, in percentage, observed in countermovement jump, medicine ball throw, handgrip, and sprint time from pre- to post-training. Mean and standard deviation are also presented in lines. ** Significant difference between experimental and control groups.

**Table 1 jfmk-10-00214-t001:** Participants’ characteristics.

	PD Group	CT Group	Group Effect (*p*-Value)
Age (years)	11.81 ± 1.25	12.22 ± 0.75	0.070
Height (m)	1.58 ± 0.04	1.60 ± 0.05	0.061
Body mass (kg)	50.00 ± 6.75	56.92 ± 2.75	0.067
BMI (kg/m^2^)	19.96 ± 1.95	21.61 ± 1.02	0.018

BMI: body mass index (kg/m^5^).

**Table 2 jfmk-10-00214-t002:** Mean and SD values of variables assessed and comparisons between pre- and post-training in padel group (PD) and control group (CT).

	Pre-Training	Post-Training	Mean Difference 95% Lower Limit–Upper Limit	* *p*	Effect Size (d)
PD					
Handgrip strength (kg)	21.68 ± 2.55	22.94 ± 2.03	−0.47 ± −2.04	* 0.004	0.55
1-kg ball throw (cm)	2.50 ± 0.26	2.60 ± 0.36	0.08 ± −0.27	0.265	0.32
T5m (s)	2.10 ± 0.33	1.95 ± 0.26	0.31 ± −0.01	0.060	0.50
CMJump (cm)	26.78 ± 5.45	26.22 ± 4.99	1.54 ± −0.43	0.250	0.11
CT					
Handgrip strength (kg)	19.85 ± 3.24	20.27 ± 1.63	1.46 ± −2.30	0.630	0.16
MBT 1-kg (m)	2.50 ± 0.21	2.52 ± 0.23	0.08 ± −0.09	0.930	0.09
T5m (s)	2.30 ± 0.19	2.45 ± 0.17	−0.04 ± −0.26	† 0.011	0.83
CMJump (cm)	24.15 ± 3.35	24.43 ± 4.09	1.32 ± −1.88	0.705	0.07

CMJump: countermovement vertical jump (m); MBT 1 kg: 1 kg medicine ball throw (m); T5m: sprint running 5 m (s); 95% CI = 95% confidence interval; * *p*-value between pre- and post-training; † *p*-value, 0.01 interaction.

## Data Availability

The data presented in this study are available on request from the corresponding author.

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
