# Peer review of "Comparative Study in Physical Fitness in Recreative Young Padel Players"

_jfmk, 2025, doi:10.3390/jfmk10020214_

Round 1
Reviewer 1 Report
Comments and Suggestions for Authors
Reviewer_ Report : Gaith Aloui
Point 1:
The purpose of this study was to compare the effect of 8-week padel training (PD) on strength and 10 power in untrained healthy children. I believe that this experimental procedure should include a lot of insightful information having useful practical suggestions for performance improvement in athletes. However, I have found some issues as shown below. The most important remark is that the part of the materials and methods and discussion part must be more detailed.
ABSTRACT
Point 2: It is better to add the characteristics of the study population
INTRODUCTION
Point 3: Line 32-34 : Enrich this paragraph by adding other physical and physiological demands of padel.
Point 4: Line 34-36 : Name the countries where this game is widespread
Point 5: Line 41-43 : More details about the padel match
Point 6: Line 44-45 : Add the age category of the study population
Point 7: Line 45-47 : Add the gender and age category of the study population.
Point 8: Line 49-52 : Enrich this paragraph by adding scientific explanations and interpretations.
Point 9: Line 55-58 : Mention whether there are recent studies that have not reported positive effects of Pabel practice on body composition, fitness capacity, and some psychosocial characteristics.
Point 10: Line 60-67 : Enrich your problem
.MATERIALS AND METHODS
Point 11 : Procedures part : Lack of details in padel training program.
Point 12 : Procedures part : Explain the choices of padel exercises.
Point 13 : Procedures part : Why did you schedule all the tests in one day.
Point 14 : Procedures part : mention if you have scheduled the tests for several days.
DISCUSSION
Summary of evidence
Point 15 : paragraph n°1 : Rephrase the paragraph n°1, it is not clear
Point 16 : paragraph n°2 : Lack of scientific explanation for improvement in running speed padel training.
Point 17 : paragraph n°2 : Discuss your results with those who have compared the effect padel training on sprint performance even in other disciplines.
Point 18 : Line253-255 : Lack of references.
Point 19 : Line260-263 : Rephrase the paragraph n°1, it is not clear
Point 20 : paragraph n°3 : Discuss your results with those who have compared the effect padel training on strength performance even in other disciplines.
Point 21 : paragraph n°3 : lack of scientific explanation for your results.
Point 22 : paragraph n°4 : It is better to summarize this paragraph, mention your ideas more clearly.
Point 23 : paragraph n°5 : It is better to summarize this paragraph mention your ideas more clearly.
Limitations
Point 24 : Please further cite the study limitations regarding the study population (number, gender, age group, etc.)
Point 24 : Mention other physical and psychomotor qualities and abilities to be assessed.
Point 25 : Mention the reduced numbers of tests in your study
Practical applications
Point 26 : well prepared, just mention the adequate dosage of padel training
CONCLUSION
Point 26 : Focus more on the age category of your study population, put more emphasis on your results.
Author Response
Reviewer_ Report : Gaith Aloui
Point 1:
The purpose of this study was to compare the effect of 8-week padel training (PD) on strength and 10 power in untrained healthy children. I believe that this experimental procedure should include a lot of insightful information having useful practical suggestions for performance improvement in athletes. However, I have found some issues as shown below. The most important remark is that the part of the materials and methods and discussion part must be more detailed.
ABSTRACT
Point 2: It is better to add the characteristics of the study population
R: thank you for all the comments, we added the information in the abstract.
INTRODUCTION
Point 3: Line 32-34 : Enrich this paragraph by adding other physical and physiological demands of padel.
R: We added the main characteristics of padel.
Point 4: Line 34-36 : Name the countries where this game is widespread
R: We added the most countries around the world.
Point 5: Line 41-43 : More details about the padel match
R: We added more game characteristic about padel match
Point 6: Line 44-45 : Add the age category of the study population
R: We added the study population
Point 7: Line 45-47 : Add the gender and age category of the study population.
R: We added the gender and age category of the study population.
Point 8: Line 49-52 : Enrich this paragraph by adding scientific explanations and interpretations.
R: We added more information according to the literature.
Point 9: Line 55-58 : Mention whether there are recent studies that have not reported positive effects of Pabel practice on body composition, fitness capacity, and some psychosocial characteristics.
R: We rewrite the lines. Because all the recent studies reported positive effects. We remove “However, the extent to which regular padel practice may benefit fitness in young players remains unknown.”
Point 10: Line 60-67 : Enrich your problem
R: We rewrite all the paragraphs because is very important to all readers.
.MATERIALS AND METHODS
Point 11 : Procedures part : Lack of details in padel training program.
R: we add information about the normal session of padel training routines
Point 12 : Procedures part : Explain the choices of padel exercises.
R: The training program were the habitual training of padel that includes volley, bandejas, using the glasses..
Point 13 : Procedures part : Why did you schedule all the tests in one day.
R: We scheduled the test for one day to avoid interfering with the participants' family logistics, as we don’t want to complicate or change the time they need to leave before and after the training session.
Point 14 : Procedures part : mention if you have scheduled the tests for several days.
R: All tests were performed in the same session, in the following order: jump, ball throwing, speed, and handgrip.
DISCUSSION
Summary of evidence
Point 15 : paragraph n°1 : Rephrase the paragraph n°1, it is not clear
R: We rewrote all the paragraphs.
Point 16 : paragraph n°2 : Lack of scientific explanation for improvement in running speed padel training.
R: We rewrite and add a great work performed by Tomás et al. (2022)
Point 17 : paragraph n°2 : Discuss your results with those who have compared the effect padel training on sprint performance even in other disciplines.
R: Thank you for the suggestions. We added:
Kramer, T.; Valente-Dos-Santos, J.; Visscher, C.; Coelho-E-Silva, M.; Huijgen, B.C.H.; Elferink-Gemser, M.T.; Longitu-dinal development of 5m sprint performance in young female tennis players. J. Sports Sci. 2021, 39(3, 296-303. DOI: 10.1080/02640414.2020.1816313
Kramer, T.; Valente-Dos-Santos, J.; Coelho-E-Silva, M.J.; Malina, R.M.; Huijgen, B.C.; Smith, J.; Elferink-Gemser, M.T.; Visscher, C. Modeling Longitudinal Changes in 5 m Sprinting Performance Among Young Male Tennis Players. Percept Mot. Skills. 2016, 122(1), 299-318. DOI: 10.1177/0031512516628367.
Point 18 : Line253-255 : Lack of references.
R: We added more references
Point 19 : Line260-263 : Rephrase the paragraph n°1, it is not clear
R: We rewrote paragraph n°1
Point 20 : paragraph n°3 : Discuss your results with those who have compared the effect padel training on strength performance even in other disciplines.
R: we added information.
Point 21 : paragraph n°3 : lack of scientific explanation for your results.
R: We rewrite; thank you.
Point 22 : paragraph n°4 : It is better to summarize this paragraph, mention your ideas more clearly.
R: we rewrite, thank you.
Point 23 : paragraph n°5 : It is better to summarize this paragraph mention your ideas more clearly.
R: we rewrite, thank you.
Limitations
Point 24 : Please further cite the study limitations regarding the study population (number, gender, age group, etc.)
R: We added the study population
Point 24 : Mention other physical and psychomotor qualities and abilities to be assessed.
R: We added some more physical and psychomotor qualities and abilities
Point 25 : Mention the reduced numbers of tests in your study
R: We added this information
Practical applications
Point 26 : well prepared, just mention the adequate dosage of padel training
R: We added this information
CONCLUSION
Point 26 : Focus more on the age category of your study population, put more emphasis on your results.
R: We focus more on the age of the participants.

Reviewer 2 Report
Comments and Suggestions for Authors
Dear Authors,
thank you for the opportunity to revise your manuscript. This paper aimed to compare the effect of 8-week Padel training on strength and power in untrained healthy children.
The main findings suggested that 8 weeks of Padel training effectively improves strength and power-related variables in healthy, untrained children.
The study is interesting, but there are some critical issues to be addressed:
TITLE
You should report your typology study on the title, for example identifying the paper as a Randomized Controlled Trial.
ABSTRACT
Please reorganize the abstract, respecting the CONSORT guidelines for reporting abstracts of Randomized Trials.
In the background of the abstract, explain the rationale of the study.
Please, clarify and describe the trial design.
In the Methods, organize and explain the eligibility criteria for participants and the settings where the data were collected.
INTRODUCTION
In the Introduction, you correctly reported the aim of the paper, but you should organize and deepen the scientific background and explanation of the rationale.
P1 L44: Please clarify in detail the level of physical fitness required to prevent injuries.
P2 L61: Please clarify how the Padel can improve asymmetries between upper and lower limbs often present in young people, despite not being a symmetrical sport.
Please improve the Introduction Section clarify the potential risk of injuries in padel players citing and discussing the following papers: doi: 10.23736/S0022-4707.23.15418-1; doi: 10.1007/s40477-023-00869-2; doi: 10.3390/ijerph19074153.
METHODS
Please, reorganize the Methods, following the CONSORT 2010 guidelines for the Randomized Controlled Trials.
Organize and clarify the trial design.
In the participant’s session, you missed the exclusion criteria.
Do you had registered your protocol on clinicaltrials.gov? Please, clarify this aspect.
You missed the settings and locations where the data were collected. Please, report it in the text.
P2 L82: The number of participants who were randomly assigned belongs to Results and not to Methods.
Please, clarify the Mechanism used to implement the random assignment.
RESULTS
In Results, create a participant flowchart with the number of participants who were randomly assigned, received the intended treatment, and were analyzed for the primary outcome.
Define the date periods of recruitment and follow-up.
Please, report Table 1, which shows baseline demographic and clinical characteristics for each group, in the Result session and not in Methods.
DISCUSSION
Although the authors have described the limitations of this study from four perspectives, it would
be helpful to additionally describe the strengths of the study that overcome its limitations in the
Discussion section.
You can also organize and deepen the external validity and applicability of the trial findings.
Author Response
Dear Authors,
thank you for the opportunity to revise your manuscript. This paper aimed to compare the effect of 8-week Padel training on strength and power in untrained healthy children.
The main findings suggested that 8 weeks of Padel training effectively improves strength and power-related variables in healthy, untrained children.
The study is interesting, but there are some critical issues to be addressed:
TITLE
You should report your typology study on the title, for example identifying the paper as a Randomized Controlled Trial.
R: The present study consists of a simple randomized controlled trial. We added information about this in the specific topics of the study and improve the abstract and also change the title.
ABSTRACT
Please reorganize the abstract, respecting the CONSORT guidelines for reporting abstracts of Randomized Trials.
R: We rewrite the description of the sample to clarify the typology of the study.
In the background of the abstract, explain the rationale of the study.
R: It was added.
Please, clarify and describe the trial design.
R: It was improved according to the study plan
In the Methods, organize and explain the eligibility criteria for participants and the settings where the data were collected.
R: We added the information
INTRODUCTION
In the Introduction, you correctly reported the aim of the paper, but you should organize and deepen the scientific background and explanation of the rationale.
P1 L44: Please clarify in detail the level of physical fitness required to prevent injuries.
R: It was added
P2 L61: Please clarify how the Padel can improve asymmetries between upper and lower limbs often present in young people, despite not being a symmetrical sport.
Please improve the Introduction Section clarify the potential risk of injuries in padel players citing and discussing the following papers: doi: 10.23736/S0022-4707.23.15418-1; doi: 10.007/s40477-023-00869-2; doi: 10.3390/ijerph19074153.
R: Thank you for the mention to recent studies. We added the work to improve our introduction and discussion.
METHODS
Please, reorganize the Methods, following the CONSORT 2010 guidelines for the Randomized Controlled Trials.
R: we clarify in the procedures the type of study
Organize and clarify the trial design.
R: We clarify the trial design non-randomized
In the participant’s session, you missed the exclusion criteria.
R: We add the information and complete the description in participants' topic
Do you had registered your protocol on clinicaltrials.gov? Please, clarify this aspect.
R: It was not registered because we do not perform a RCT
You missed the settings and locations where the data were collected. Please, report it in the text.
R: We add the region of Portugal
P2 L82: The number of participants who were randomly assigned belongs to Results and not to Methods.
R: Thank you It was changed
Please, clarify the Mechanism used to implement the random assignment.
R: It was added in the procedures
RESULTS
In Results, create a participant flowchart with the number of participants who were randomly assigned, received the intended treatment, and were analyzed for the primary outcome.
R: We added the information to the results. We performed a flowchart, but we think that in the present study, it will not seem to be appropriate. But we did it:
Assessed for eligibility (n=25) |
Enrollment |
Non-Randomized (n=25) |
Excluded (n=0)
|
Allocated to Padel Group (n=14) -Training Padel |
Allocated to Control Group (n=11) -Do not performed intervention |
Allocation |
Number of participants lost during the 8 weeks (n=0)
|
Analyzed (n=11)
|
After 8-week |
Analysis |
Number of participants lost during training (n=0)
|
Analyzed (n=14) · |
Padel Group (n=14)
|
Control Group (n=11)
|
Define the date periods of recruitment and follow-up.
R: it was added
Please, report Table 1, which shows baseline demographic and clinical characteristics for each group, in the Result session and not in Methods.
R: Thank you, it changed
DISCUSSION
Although the authors have described the limitations of this study from four perspectives, it would be helpful to additionally describe the strengths of the study that overcome its limitations in the Discussion section
You can also organize and deepen the external validity and applicability of the trial findings.
R: Thank you for all the suggestions. We improved the discussion part

Reviewer 3 Report
Comments and Suggestions for Authors
Comments can be read in the submitted document.

Author Response
Revisor PDF
Twenty-five children aged 11–15 years (mean SD: 12.36 1.15 11 years) were randomly assigned into experimental (PD) and control (CT) groups. PD group (9 boys and 5 girls) trained padel twice a week for 8 weeks, and CT (7 boys and 4 girls) did not follow any training program.
A:Abstract Rewritted
It is recommended to replace this expression, "differences between children who don’t perform any sport", with the following one: "compared to children who do not engage in any sports". activities.
A: Rewritted
In the age range of these samples, there is significant variability in the maturity status of individuals, particularly in terms of sexual maturity and hormone levels, which affect muscle mass. If possible, although the authors may not have this data, the effect of the interaction due to the sexual maturity status of each participant could be analyzed. This could be achieved using Tanner staging, even if self-reported.
A: We agree but we did not collect that data.
This difference in body mass index (BMI) between the two groups should be considered when discussing the results. A priori, and considering the younger children aged 11 to 12 years, those in the experimental group could be classified within a normal BMI category, whereas those in the control group might fall into an overweight category. This discrepancy could potentially influence the manifestation of other physical fitness variables.
A: Thank you for the mention. We add this also in the limitations of our study because we do not measure muscle mass, adiposity, water, …
This information regarding informed consent is repeated. It is suggested to rephrase it to avoid redundancy.
A: We rewritted.
This description requires additional information to enable the replication of the test. It is not specified whether the start is voluntary or initiated by a command from the evaluator. In the latter case, the reaction time of each individual would need to be considered, not just the speed they can achieve. Additionally, it is unclear how the measurement is taken, where the photoelectric cells are positioned at the start, and whether the individual begins from a stationary position or a running start. Furthermore, the number of attempts each person is allowed is not mentioned.
A: The test was supported with reference but we add a more detailled discription.
It is suggested to revise the wording of this sentence because positive responses to the training can only occur in the experimental group, never in the control group.
A: We agree. We rewritted.

Reviewer 4 Report
Comments and Suggestions for Authors
The manuscript addresses a relevant topic by investigating the effects of padel training on recreational young athletes, contributing to a little-explored field. The methodology is well-structured, with appropriate instruments and promising results in handgrip strength and speed. However, the small sample size, the lack of control for differences in biological maturity, and a limited analysis of variables with no significant impact (e.g., vertical jump) reduce the study's robustness. Additionally, the integration of results with existing literature could be broader, including comparisons with other sports.
Title
The title adequately reflects the study's theme, but it could be more descriptive. I suggest including "comparative study" to highlight the focus on group comparison.
Abstract
The abstract presents the main results but does not describe them clearly enough. I recommend including numerical values for the most significant variables to provide greater context and revising inconsistencies, such as the formatting of "np2 p=0.31." It should be adjusted to "η² = 0.31" to follow the standard notation in statistics.
Line 35: Replace "it has also been growing in popularity and participation" with "it has been growing in popularity," simplifying the sentence.
Lines 36-37: Replace "increasing the need to increase knowledge" with "highlighting the need to expand knowledge."
Introduction
Present the hypothesis directly. For example: "We hypothesize that 8 weeks of padel training will result in significant improvements in strength and speed compared to the control group."
Methods
Detail how randomization was conducted. For example: "simple randomization using software."
Table 1: Add descriptions of the abbreviated variables and adjust the formatting to separate the standard deviation symbol (e.g., ±) from the numbers.
Was any dietary control conducted for the sample? If not, this should be included in the limitations section.
For example, consistently use "players" instead of sometimes writing "participants" and other times writing "players."
Line 149: Add the reference.
I suggest calculating the sample size using G*Power.
Lines 179-185: Combine the information into a single paragraph.
Line 182: Remove the quotation marks from "Group x Pre-Post."
Lines 186-187: "Practice experience may affect BM in both groups" is ambiguous. I recommend rewriting it as "Practice experience might have influenced the performance in the MBT for both groups."
Table 2: Some units appear with one decimal place and others with two; maintain consistency. Also, separate the standard deviation sign from the numbers. Add a symbol highlighting statistical differences.
Figure 1 Legend: Remove abbreviations.
The discussion aligns well with previous findings, but it lacks depth in exploring the reasons why certain variables were affected by the training while others were not. Additionally, comparisons between the impacts of padel training and other racket sports (e.g., tennis, badminton) are missing.
Line 250: Change "We" to "we."
Line 255: Change "youngsters may lag in play and become isolated" to "Young individuals may experience reduced physical engagement and social interaction."
Line 299: Delete (REF).
Line 344: The phrase "Informed consent was obtained from all subjects involved in the study" could include: "and their guardians," since the participants are minors.
Author Response
The manuscript addresses a relevant topic by investigating the effects of padel training on recreational young athletes, contributing to a little-explored field. The methodology is well-structured, with appropriate instruments and promising results in handgrip strength and speed. However, the small sample size, the lack of control for differences in biological maturity, and a limited analysis of variables with no significant impact (e.g., vertical jump) reduce the study's robustness. Additionally, the integration of results with existing literature could be broader, including comparisons with other sports.
Title
The title adequately reflects the study's theme, but it could be more descriptive. I suggest including "comparative study" to highlight the focus on group comparison.
R: Dear Reviewer, thank you for all the comments. We want to improve this field in Padel sport. We change the title to increase the highlights in sport science.
Abstract
The abstract presents the main results but does not describe them clearly enough. I recommend including numerical values for the most significant variables to provide greater context and revising inconsistencies, such as the formatting of "np2 p=0.31." It should be adjusted to "η² = 0.31" to follow the standard notation in statistics.
R: Thank you for the correction, we add and correct the information.
Line 35: Replace "it has also been growing in popularity and participation" with "it has been growing in popularity," simplifying the sentence.
R: Thank you for the correction, we add and correct the information.
Lines 36-37: Replace "increasing the need to increase knowledge" with "highlighting the need to expand knowledge."
R: Thank you for the correction, we add and correct the information.
Introduction
Present the hypothesis directly. For example: "We hypothesize that 8 weeks of padel training will result in significant improvements in strength and speed compared to the control group."
R: Thank you for the correction, we add and correct the information.
Methods
Detail how randomization was conducted. For example: "simple randomization using software."
R: Dear reviewer, we added it. Thank you
Table 1: Add descriptions of the abbreviated variables and adjust the formatting to separate the standard deviation symbol (e.g., ±) from the numbers.
R: Thank you for the correction, we add and correct the information.
Was any dietary control conducted for the sample? If not, this should be included in the limitations section.
R: In this case we don’t perform it. We add the information in the limitations section.
For example, consistently use "players" instead of sometimes writing "participants" and other times writing "players."
R: We performed a revision in the terms used and correct it according to the designation in the text.
Line 149: Add the reference.
R: We add the reference.
I suggest calculating the sample size using G*Power.
R: Dear reviewer, we do not perform it because the sample size is small, but the participants were what we achieved in these ages.
Lines 179-185: Combine the information into a single paragraph.
R: We rewrite the paragraph.
Line 182: Remove the quotation marks from "Group x Pre-Post."
R: Thank you for the correction, we add and correct the information.
Lines 186-187: "Practice experience may affect BM in both groups" is ambiguous. I recommend rewriting it as "Practice experience might have influenced the performance in the MBT for both groups."
R: Thank you for the correction, we add and correct the information.
Table 2: Some units appear with one decimal place and others with two; maintain consistency. Also, separate the standard deviation sign from the numbers. Add a symbol highlighting statistical differences.
R: Thank you for the correction, we add and correct the information.
Figure 1 Legend: Remove abbreviations.
R: Thank you for the correction, we correct the information.
The discussion aligns well with previous findings, but it lacks depth in exploring the reasons why certain variables were affected by the training while others were not. Additionally, comparisons between the impacts of padel training and other racket sports (e.g., tennis, badminton) are missing.
Line 250: Change "We" to "we."
R: Thank you for the correction, we correct the information.
Line 255: Change "youngsters may lag in play and become isolated" to "Young individuals may experience reduced physical engagement and social interaction."
R: Thank you for the correction, we correct the information.
Line 299: Delete (REF).
R: Thank you for the correction, we delete the information.
Line 344: The phrase "Informed consent was obtained from all subjects involved in the study" could include: "and their guardians," since the participants are minors.
R: Thank you for the correction, we add the information.

Round 2
Reviewer 1 Report
Comments and Suggestions for Authors
All points have been carefully and thoroughly reviewed by the author.
Author Response
Thank you a lot for the revision. We will increase the future studies and try to present new contributions in Padel, to prevent injuries and also to promote the training of this sport in schools. In both genders.
Reviewer 2 Report
Comments and Suggestions for Authors
Dear Authors,
I would like to congratulate with you for having improved your paper.
Please just improve the background and the study rationale also discussing the risk of injury in padel players. Accordingly, you can cite and discuss the following papers:
doi: 10.23736/S0022-4707.23.15418-1.
doi: 10.1080/14763141.2025.2468320
Comments on the Quality of English LanguagePlease revise the entire paper to improve the Quality of English Language.
Author Response
Dear Authors,
I would like to congratulate with you for having improved your paper.
Please just improve the background and the study rationale also discussing the risk of injury in padel players. Accordingly, you can cite and discuss the following papers:
doi: 10.23736/S0022-4707.23.15418-1.
doi: 10.1080/14763141.2025.2468320
-----
R: Dear reviewer, we add a new paragraph about injuries in padel sport. It is very important to perform a kinematic analysis in young players also, because we don’t study it but we observe during games, competitive level, some players using support materials to prevent pain.
- Viera, H.L.S.; Leite-Nunes, T.D.; Gidiel-Machado, L.; Laporta, L.I.; Royes, L.F.F.; Forgiarini Saccol, M.; Lanferdini, F.J. Assessment of shoulder joint and muscle characteristics side-asymmetry in professional padel players. Sports Biomech 2025, 24, 1-17. doi:10.1080/14763141.2025.2468320
- de Sire, A.; Demeco, A.; Frizziero, A.; Marotta, N.; Spanò, R.; Carozzo, S.; Costantino, C.; Ammendolia, A. Risk of injury and kinematic assessment of the shoulder biomechanics during strokes in padel players: a cross-sectional study. J Sports Med Phys Fitness 2024, 64, 383-391. doi: 10.23736/S0022-4707.23.15418-1

Reviewer 4 Report
Comments and Suggestions for Authors
None.
Author Response
Thank you a lot for the revision. We will improve the scientific research in this field and try to contribute with new ideas about the practice of padel in both genders in young participants.